# Antimicrobial Efficacy of Five Probiotic Strains Against *Helicobacter pylori*

**DOI:** 10.3390/antibiotics9050244

**Published:** 2020-05-11

**Authors:** Ilaria Maria Saracino, Matteo Pavoni, Laura Saccomanno, Giulia Fiorini, Valeria Pesci, Claudio Foschi, Giulia Piccirilli, Giulia Bernardini, John Holton, Natale Figura, Tiziana Lazzarotto, Claudio Borghi, Berardino Vaira

**Affiliations:** 1Department of Surgical and Medical Sciences, University of Bologna, 40138 Bologna, Italy; saracinoilariamaria@gmail.com (I.M.S.); matteo.pavoni@studio.unibo.it (M.P.); laura.saccomanno@studio.unibo.it (L.S.); giulia.fiorini@fastwebnet.it (G.F.); valeria.pesci@gmail.com (V.P.); claudio.borghi@unibo.it (C.B.); 2Microbiology and Clinical Microbiology, Department of Experimental, Diagnostic and Specialty Medicine, University of Bologna, 40138 Bologna, Italy; claudio.foschi2@unibo.it (C.F.); giulia.piccirilli2@unibo.it (G.P.); tiziana.lazzarotto@unibo.it (T.L.); 3Department of Biotechnology Chemistry and Pharmacy, University of Siena, 53100 Siena, Italy; bernardini@unisi.it (G.B.); figuranatale@gmail.com (N.F.); 4Department of Health & Social Sciences, University of Middlesex, London NW4 4HE, UK; john.holton@nhs.net

**Keywords:** probiotics, *Helicobacter pylori*, dysbiosis, therapy, bactericidal activity, bacteriostatic activity

## Abstract

Treatment of *Helicobacter pylori* (*H. pylori*) infection is a challenge for clinicians. The large increase in drug-resistant strains makes the formulation of new therapeutic strategies fundamental. The frequent onset of side effects during antibiotic treatment (mainly due to intestinal dysbiosis) should not be underestimated as it may cause the interruption of treatment, failure of *H. pylori* eradication and clonal selection of resistant bacteria. Probiotic integration during antibiotic treatment can exert a dual function: a direct antagonistic effect on *H. pylori* and a balancing effect on dysbiosis. Therefore, it fulfills the definition of a new therapeutic strategy to successfully treat *H. pylori* infection. Data reported in literature give promising but discrepant results. Aim: To assess in vitro bacteriostatic and bactericidal activity of probiotic strains against *H. pylori*. Materials and methods: *L. casei, L. paracasei, L. acidophilus, B. lactis* and *S. thermophilus* strains were used. Agar well diffusion and time-kill curves were carried out to detect bacteriostatic and bactericidal activity, respectively. Results: All probiotic strains showed both bacteriostatic and bactericidal activity vs. *H. pylori*. Conclusions: Such findings prompted us to plan a protocol of treatment in which probiotics are given to infected patients in association with antibiotic therapy.

## 1. Introduction

*Helicobacter pylori* is a Gram-negative microaerophilic bacterium that colonizes the gastric mucosa, causing gastritis and peptic ulcers. It leads to the development of gastric-mucosa-associated lymphoid tissue (MALT) lymphoma and gastric carcinoma [1]. Treatment of *H. pylori* infection is a challenge for clinicians and the large increase in drug-resistant strains globally makes the development of new therapeutic approaches crucial [2]. Additionally, not to be underestimated is the clinical significance of frequent adverse events caused by current antibiotic treatments (mainly due to intestinal dysbiosis). These adverse events result in interruption of therapy, favoring the clonal selection of resistant strains and creating a vicious cycle [3].

Several bacterial *Phyla* have been identified in the stomach. The gastric microbiota is dynamic and is affected by several factors, establishing multiple interactions with the gastric mucosa and, when present, with *H. pylori*. Overall, in the presence of *H. pylori,* a loss of microbiota biodiversity occurs, leading to dysbiosis. Probiotic use for an *H. pylori* infected stomach has many beneficial immunological and non-immunological effects: it enables the rebalancing of pro- and anti-inflammatory cytokines, activates defense mechanisms against pathogens and improves the mucosal barrier. Furthermore, probiotics compete with *H. pylori* for the same adhesion sites and nutrients and induce the production of mucin; they also produce metabolites with antimicrobial activity [4,5]. Probiotic integration during antibiotic therapies can also balance intestinal dysbiosis, thus decreasing dysbiosis-induced adverse events and increasing patient compliance. For these reasons, integration with probiotics fully responds to the need of new therapeutic cues to successfully treat *H. pylori* infection. A recent meta-analysis stated that “probiotics improved the eradication rate and reduced side effects when added to the treatments designed to eradicate *H pylori*. The use of probiotics either before and throughout the eradication treatment, exerted better eradication effects” [6]. However, data reported in the literature give discrepant results. The aim of this study was to assess in vitro the bacteriostatic and bactericidal activity of probiotic strains against *H. pylori*, as a preliminary experiment to plan a clinical trial.

## 2. Results 

Viable counts (range, mean, standard deviation) and pH ranges of probiotic strains after overnight culture are reported in Table 1. It was not possible to normalize the quantity of cells after overnight cultures due to the different bacterial growth rates. Overnight broth-cultures were not diluted in order not to modify their metabolite content.

Fifty-seven *H. pylori* strains, grouped according to the eight resistance patterns observed, were tested (Table 2). Antibiotic susceptibility tests were performed with the E-test method.

All probiotic strains showed bacteriostatic activity against *H. pylori* (Figure 1). All the five strains generated inhibition zones (IZs) larger than those of the respective negative controls (MRS (DeMan-Rogosa-Sharpe) and BHI (brain heart infusion) broths) (*p* < 0.05). IZs obtained with *L. casei, L. paracasei* and *L. acidophilus* were very similar to each other and were greater than those generated by *B. lactis*, which in turn created larger IZs than those generated by *S. thermophilus*. These differences were statistically significant (*p* < 0.05). IZ means and their standard errors are reported in Figure 1.

No differences were observed in the IZ means of the same probiotic strain vs. the eight *H. pylori* groups. This confirms that *H. pylori* antibiotic resistance mechanisms do not interfere with its susceptibility to the antimicrobial activity of probiotic strains (Figure 2).

*L. casei, L. paracasei* and *L. acidophilus* inhibited all *H. pylori* strains; *B. lactis* inhibited 89.5% (51/57) of them; *S. thermophilus* inhibited 18% (10/57) of them but its IZs, when present, were large. 

IZ range for *L. casei, L. paracasei, L. acidophilus, B. lactis* and *S. thermophilus* were, respectively, 4 mm to 14 mm, 4 mm to 15 mm, 4 mm to 14 mm, no inhibition to 10 mm and no inhibition to 23 mm. In Figure 3, we report the *H. pylori* growth inhibition caused by *Lactobacillus* spp.

A time-kill study was performed on a multi-resistant *H. pylori* strain. At T0, T3, T6, T24 and T72 h, *H. pylori* total viable count was performed. Results are reported in Figure 4.

At 3 h, *H. pylori* aliquots co-incubated with *L. acidophilus, L. casei, L. paracasei* and *B. lactis* supernatants showed a decrease of 10^3^ CFU/mL ca. The aliquot co-incubated with *S. thermophilus* supernatant showed a decrease of approximately 10^2^ CFU/mL. The *H. pylori* viable count continued to decrease steadily until there were no viable cells at 72 h. Viable count of the negative control doubled after 72 h of incubation. 

## 3. Discussion

Research on the gastric microbiota is a recent topic [7], as a consequence of the idea that “the stomach is a sterile organ”, inhospitable to bacteria. This short-sighted principle arose because of low gastric pH, duodeno-gastric reflux of bile, thickness of the mucus layer and gastric peristalsis; these are all factors suggesting that bacteria could not exist in such an environment. On the contrary, several *Phyla* have been identified in the stomach, some present in the gastric lumen (transient colonies) and others colonizing the mucosa (stable colonies). The gastric microbiota is complex and dynamic; it establishes multiple interactions with the mucosa and, when present, also with *H. pylori* [8,9,10]. Probiotic bacteria can inhibit *H. pylori* through immunological and/or non-immunological mechanisms. It is known that *H. pylori* infection induces the production of nuclear factor-kB (NF-kB), interleukin-8 (IL-8) and tumor necrosis factor α (TNFα). IL-8 leads to the migration of neutrophils and monocytes into the mucosa; the activated monocytes and dendritic cells stimulate the production of various cytokines, such as IL-4, IL-5, IL-6 and interferon-*γ* (IFN-γ), leading to an inflammatory reaction (chronic gastritis). The persistence of the inflammatory insult is associated with the development of gastric cancer [11]. Probiotics can modify the immunologic response of the host by interacting with epithelial cells and modulating the secretion of anti-inflammatory cytokines and chemokines, resulting in a reduction of gastric inflammation, mainly through the inhibition of the NF-kB pathway [12,13]. Furthermore, probiotics can enhance the production of secretory IgA, an additional defense against pathogens [14]. Non-immunological mechanisms of probiotics include the production of antimicrobial substances (bacteriocins), inhibition of adherence to the gastric mucosa, stimulation of mucin production and stabilization of the gut mucosal barrier. Furthermore, lactic acid bacteria and bifidobacteria produce organic acids, hydrogen peroxide, carbon dioxide and other antimicrobial compounds that may inhibit potential pathogens. Adhesion of pathogens can be inhibited either by co-aggregation and steric hindrance, or by competing for specific carbohydrate receptors [4,15,16,17].

*H. pylori* causes chronic gastritis, peptic ulceration, and may lead to gastric MALT lymphoma and carcinoma development [1]. Treatment of infection remains a challenge for clinicians; in fact, no therapeutic protocol is 100% effective. What makes the formulation of new therapies fundamental is the substantial increase in drug-resistant strains [2]. In this situation, the role played by the frequent onset of adverse reactions during antibiotic therapies should not be underestimated. They are mainly due to intestinal dysbiosis and consist of the appearance of nausea, vomiting, diarrhea, etc. [3]. These factors notably decrease the compliance of patients to therapy, creating a vicious cycle between adverse reactions, suspension of therapy, clonal selection of resistant bacteria and failure of subsequent therapy. In a recent study, our group observed a drastic increase in resistance rates to all antibiotics tested in patients with at least one failed therapeutic attempt [18]. Given this clear evidence, integration of probiotic strains before and during antibiotic therapy should increase the eradication rates and decrease the onset of dysbiosis-induced adverse events. Although the Maastricht Consensus states that “probiotics associated with antibiotic therapy have positive effects on the management of *H. pylori* infection” [19], meta-analyses of clinical trials published in this field give discrepant results, clearly supporting the need for further research in this area [20,21,22,23]. It is not possible to compare published clinical trials because of the extreme variability in terms of probiotic formulations used (type and duration of integration) and antibiotic therapies administered. The same discrepancies can be observed in in vitro studies: some researchers detected different levels of bacteriostatic activity in different probiotic strains [24,25,26,27], while others did not detect any inhibition at all [28]. For these reasons, the aim of this study was to assess the in vitro bacteriostatic and bactericidal activity of probiotic strains against *H. pylori*, as a preliminary investigation to set up an effective clinical trial. Probiotic strains were selected for their resistance to low pH and bile salts [29,30,31] and their growth characteristics. *Actinobacteria* (*Bifidobacterium* spp.) create transient colonies, which fluctuate in the gastric lumen and pass into the intestine, while *Firmicutes* (*Lactobacillus* spp. and *Streptococcus* spp.) can form stable colonies in the gastric mucosa [8]. Therefore, we hypothesized that *Lactobacillus* spp. and *S. thermophilus* can act directly in the stomach, while *B. lactis* may work in the intestine, preventing dysbiosis. Furthermore, as noted above, the bactericidal activity of probiotics against *H. pylori* is partially due to bacteriocins. In our opinion, since their identity is not currently fully recognized, it is preferable to use a wide range of bacteriocins, instead of those produced only by one organism, as supplement for *H. pylori* infection treatment. All probiotic strains used showed bacteriostatic and bactericidal activities against *H. pylori*. Specifically, *L. casei, L. paracasei* and *L. acidophilus* showed the best performance since they inhibited 100% of *H. pylori* strains, regardless of their pattern of antibiotic resistance. *S. thermophilus* inhibited only 18% of *H. pylori* strains, but, when present, inhibition halos were among the widest ones (up to 23 mm).

Antibiotic resistance mechanisms of *H. pylori* are not fully understood [32]. The known ones should have nothing to do with the mechanisms that make *H. pylori* susceptible to the antimicrobial action of probiotics (at least in vitro, i.e., low pH and inhibition of urease activity). However, because data on resistance patterns were available, we thought it would be wiser to divide the statistics by “*H. pylori* resistance patterns”, in order to observe if any changes in the efficacy of probiotics were detectable.

Focusing on bactericidal activity, the most interesting information is the drastic decrease of *H. pylori* viable count in the first three hours of incubation: viable cells decreased from 10^8^ to 10^5^ per mL *ca* when co-incubated with *Lactobacillus* spp. and *B. lactis* supernatants, and to 10^6^ per mL ca. when co-incubated with *S. thermophilus* supernatants.

It was not possible to normalize the quantity of cells after overnight culture, due to the different bacterial growth rates. Overnight broth-cultures were not diluted in order not to modify their content of metabolites. For homofermentative bacteria, the antimicrobial activity detectable in vitro is mainly due to two factors: the low pH and lactic acid itself, which may inhibit *H. pylori* urease enzyme [27]. *B. lactis* and *S. thermophilus* growth rates were lower than those of *Lactobacillus* spp., but, while *B. lactis* supernatant was as acidic as *Lactobacillus* spp. supernatant, *S. thermophilus* produced a supernatant with a higher pH. These observations may explain the reason why not all *H. pylori* strains were inhibited by *B. lactis* and *S. thermophilus* in the agar well diffusion test. We can assume that the same two factors (low pH and lactic acid content) played partial roles in the bactericidal activity assay; in fact, *S. thermophilus* bactericidal activity was the least effective against *H. pylori* in the first 3 h of co-incubation.

## 4. Materials and Methods 

### 4.1. Helicobacter Pylori Culture and Susceptibility Test

Fifty-seven *H. pylori* clinical isolates were cultured on the commercial selective medium Pylori Agar (BioMérieux S.p.A. Florence, Italy). The plates were incubated in jars under microaerobic conditions (CampyGen GasPack, Oxoid S.p.A Milan, Italy) at 37 °C for 3 to 5 days. The colonies resembling *H. pylori* were identified by Gram stain and oxidase, catalase and urease tests. *H. pylori* colonies were then suspended in sterile saline solution at a density corresponding to McFarland opacity standard #4 (1 McF = 3 × 10^8^ cells/mL) to perform antibiotic susceptibility test vs. clarithromycin, metronidazole and levofloxacin. The *E-Test* method was used as follows: a total of 4 agar plates for every *H. pylori* strain were streaked in 3 directions with a swab dipped into each bacterial suspension to produce a lawn of growth. Three *E-Test* strips (BioMérieux S.p.A. Florence, Italy) were placed onto three plates (one strip per plate), which were incubated immediately in a microaerobic atmosphere at 37 °C for 72 h. A fourth plate was used as positive control of bacterial development. Clarithromycin, metronidazole and levofloxacin resistance break points for the minimal inhibitory concentration (MIC) were: greater than 0.5 mg/L, greater than 8 mg/L and greater than 1 mg/L, respectively, according to the updated recommendations of the European Committee on Antimicrobial Susceptibility Testing (EUCAST 2020) [33]. All tests were carried out in duplicate.

### 4.2. Probiotic Strains

*Lactobacillus casei* DGDG (Sofar S.p.A., Milan, Italy), *Lactobacillus paracasei* LPC-S01 (Sofar S.p.A., Milan, Italy), *Lactobacillus acidophilus* LA14 (Danisco S.p.A. Milan, Italy), *Bifidobacterium lactis* BL04 (Danisco S.p.A. Milan, Italy), *Streptococcus thermophilus* ST21 (Danisco S.p.A., Milan, Italy) were tested. Lyophilized *L. casei, L. paracasei, L. acidophilus* and *B. lactis* were suspended in saline solution and seeded on DeMan-Rogosa-Sharpe (MRS) agar (Oxoid S.p.A.,Milan, Italy) with 0.05% L-cysteine, incubated in jar for 24 h at 37 °C in an anaerobic atmosphere (AnaeroGen GasPack, Oxoid S.p.A., Milan, Italy). Lyophilized *S. thermophilus* was suspended in saline solution and seeded onto blood agar containing 5% horse blood (HB) (Kima S.r.L., Padova, Italy), incubated in jars for 24 h at 37 °C under microaerobic conditions (CO2Gen GasPack, Oxoid S.p.A., Milan, Italy). To confirm bacterial species identification, matrix-assisted laser desorption ionization-time of flight mass spectrometry (Maldi-Tof microflex, Bruker Daltonics S.R.L., Macerata, Italy – SciLsLabSoftware, 3D version 2016b (Bruker Daltonics S.r.L, Macerata, Italy) was performed [34].

### 4.3. Overnight Broth Cultures 

Two colonies of *L. casei, L. paracasei, L. acidophilus* and *B. lactis* were suspended in 10 mL of MRS broth. Two colonies of *S. thermophilus* were suspended in 10 mL of brain heart infusion (BHI) broth (Oxoid S.p.A., Milan, Italy). Suspensions were kept for 20 h under specific culture conditions (24 h at 37 °C in anaerobiosis for *Lactobacillus* spp. and *B. lactis*, 24 h at 37 °C in CO_2_ enriched atmosphere for *S. thermophilus*).

### 4.4. Supernatants

Overnight cultures were centrifuged at 2500× *g* for 15 min and supernatants were filtered through 0.22 μm pore size membranes. 

### 4.5. Agar Well Diffusion (Bacteriostatic Activity). 

Fifty-seven clinical *H. pylori* isolates were used. Each strain was suspended in saline solution at a density corresponding to McFarland opacity standard #2 and seeded onto Mueller–Hinto fastidious (MHF) agar (Kima S.r.L, Padova, Italy) using a sterile swab. With the aid of a sterile tip, 7 wells (7 mm diameter) were drilled into the agar. Then, 100 μL of each overnight culture was deposited into 5 wells and 100 μL of plane broth was deposited into 2 wells for negative control. Growth inhibition zones (IZs) were read after 72 h of incubation at 37 °C in microaerobic environment. IZs were reported as the diameters of growth inhibition (well diameter was subtracted). Agar well diffusion method was carried out following the scheme reported in Figure 5.

### 4.6. Time-Kill Curve (Bactericidal Activity)

One *H. pylori* strain resistant to clarithromycin, metronidazole and levofloxacin was co-incubated with the supernatant (derived from overnight culture) of each one of the five probiotic strains. The *H. pylori* strain was suspended in modified Brucella broth (MBB) at a density corresponding to McFarland opacity standard #1, then the suspension was divided in 6 aliquots. A total of 5 were co-incubated with probiotic supernatants (1:2 *v*/*v*); the sixth aliquot was diluted 1:2 with MBB and was used as a negative control. The mixtures were thereafter incubated at 37 °C in jars under microaerobic conditions (jars were placed on a stirrer). At T0-3-6-24-72 h, scalar dilutions (1:10) were performed to determine the viable count. From each dilution, 100 µL was subcultured onto agar plates that were incubated in microaerobiosis at 37 °C for 3–5 days. Plates were then inspected and the colonies were counted; their number was expressed as colony forming unit (CFU) per mL. A time-kill curve was carried out following the scheme reported in Figure 6. All tests were performed in duplicate.

### 4.7. Statistical Analysis 

Student’s *t*-test and ANOVA were used for comparison between independent samples. A *p* value less than 0.05 was considered significant. Statistical analysis was performed with MedCalc19.1.

## 5. Conclusions

All five considered probiotic strains showed both bacteriostatic and bactericidal activity against *H. pylori* in vitro. *L. casei, L. paracasei* and *L. acidophilus* were the most effective in both tests. These results must be validated in vivo by a randomized clinical trial, which will enable us to assess the putative increase of eradication rates and the potential decrease of adverse events. The correct use of probiotics as adjuvants in antibiotic therapy against *H. pylori* could represent a turning point in the management of *H. pylori* positive patients, especially in cases of multidrug resistance.

## Figures and Tables

**Figure 1 antibiotics-09-00244-f001:**
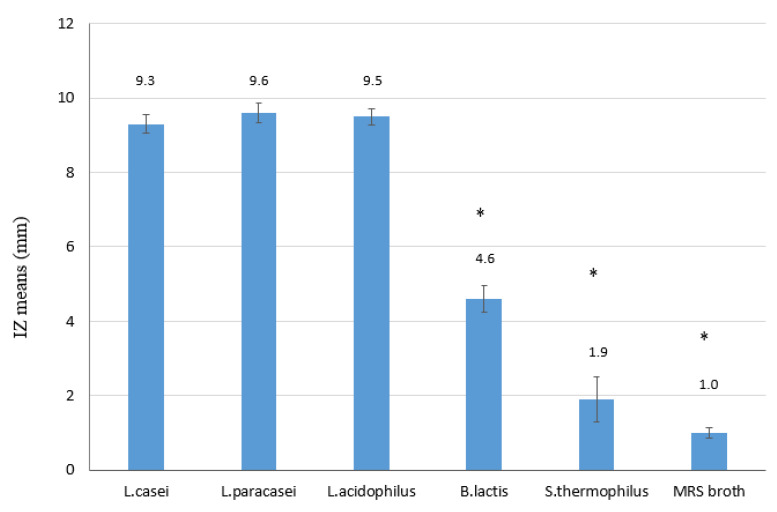
Inhibition zone (IZ) means of the five probiotic strains against *H. pylori*. DeMan-Rogosa-Sharpe (MRS) and Brain Heart Infusion (BHI) broths were used as negative controls. BHI broth never produced an IZ. IZs trend: *L. casei* = *L. paracasei* = *L. acidophilus* > *B. lactis* > *S. thermophiles* > negative control. * *p* < 0.05 was considered significant.

**Figure 2 antibiotics-09-00244-f002:**
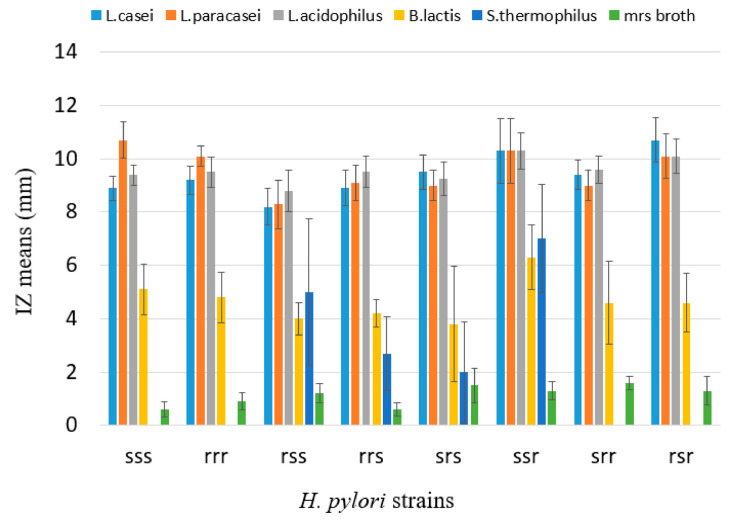
IZ means (and standard errors) in the eight *H. pylori* groups with different resistance patterns. S: susceptible. R: resistant. Resistance patterns are reported in this order: clarithromycin, metronidazole, levofloxacin. BHI broth never produced an IZ. IZs do not correlate with *H. pylori* pattern of antibiotic resistance. IZ distributions and means of the same probiotic strain vs. the eight different *H. pylori* groups were analyzed using ANOVA and Student’s *t*-test; no statistically significant difference was detected. Differences of IZ means were still significant between the probiotic strains as previously reported: *L. casei* = *L. paracasei* = *L. acidophilus* > *B. lactis* > *S. thermophiles* > negative cntr.

**Figure 3 antibiotics-09-00244-f003:**
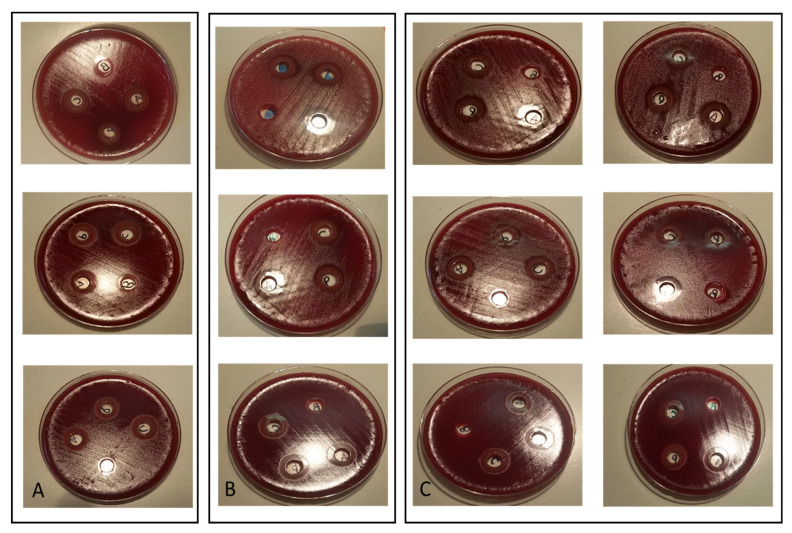
IZs from *L. casei*, *L. paracasei*, *L. acidophilus* (negative control, MRS broth). *H. pylori* susceptibility test was carried out vs. clarithromycin, metronidazole and levofloxacin. (**A**) *H. pylori* strains susceptible to all antibiotics. (**B**) *H. pylori* strains resistant to all antibiotics. (**C**) *H. pylori* strains resistant to one or two antibiotics. IZs are clearly visible.

**Figure 4 antibiotics-09-00244-f004:**
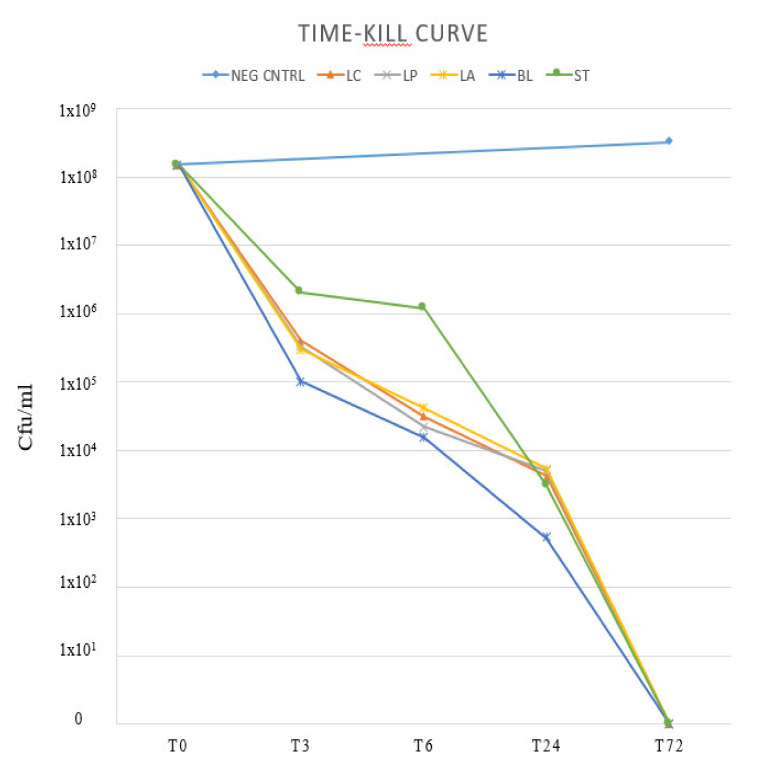
Time-kill curve. One multi-resistant *H. pylori* strain was co-incubated with supernatants derived from overnight broth culture of each probiotic strain. After only 3 h of incubation, *H. pylori* viable cell count decreased from 10^8^ to 10^6^–10^5^ CFU/mL: CFU: colony-forming units. LC: *L. casei*. LP: *L. paracasei*. LA: *L. acidophilus*. BL: *B. lactis*. ST: *S. thermophilus*.

**Figure 5 antibiotics-09-00244-f005:**
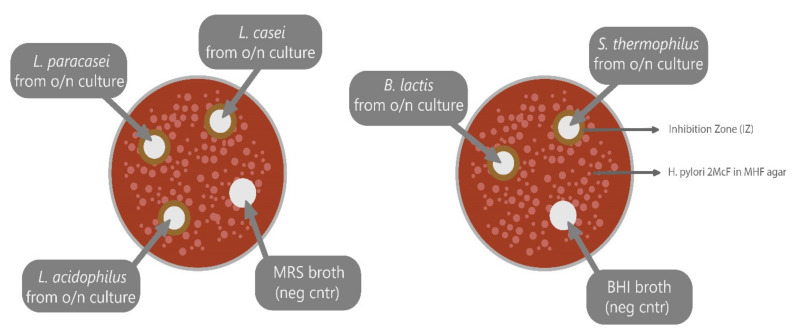
Agar well diffusion seeding scheme. o/n: overnight. McF: McFarland. MHF: Mueller-Hinton-Fastidious agar.

**Figure 6 antibiotics-09-00244-f006:**
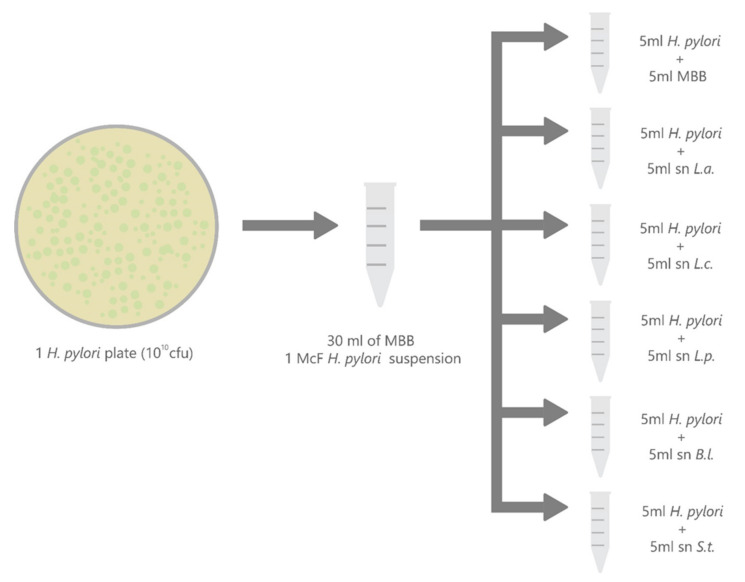
Aliquots of *H. pylori* suspension co-incubated with supernatants (sn) of probiotic strains (1:2 *v*/*v*). CFU: colony forming unit. MBB: modified brucella broth. L.a.: *L. acidophilus*. L.c.: *L. casei*. L.p.: *l. paracasei*. B.l.: *B. lactis*. S.t.: *S. thermophilus*.

**Table 1 antibiotics-09-00244-t001:** Cells/mL and pH range after 20-h overnight culture.

Probiotic Strain	N° of Cells/mL (Range)	Means	SD	pH Range
*L. casei*	9 × 10^9^–1 × 10^10^	9.4 × 10^9^	3.1 × 10^8^	4 to 4.5
*L. paracasei*	6.3 × 10^9^–1 × 10^10^	7.7 × 10^9^	1 × 10^9^	4 to 4.5
*L. acidophilus*	6.6 × 10^9^–8 × 10^9^	7.1 × 10^9^	4.5 × 10^8^	4 to 4.5
*B. lactis*	3.6 × 10^9^–7.5 × 10^9^	6.1 × 10^9^	1.8 × 10^9^	4 to 4.5
*S. thermophilus*	3 × 10^9^–6.9 × 10^9^	3.7 × 10^9^	1.4 × 10^9^	5 to 6

SD: standard deviation.

**Table 2 antibiotics-09-00244-t002:** *H. pylori* strains tested with agar well diffusion.

Resistance Patterns	Number of Strains
ClaS, MzS, LS	9
ClaR, MzR, LR	10
ClaR, MzS, LS	9
ClaR, MzR, LS	10
ClaS, MzR, LS	4
ClaS, MzS, LR	3
ClaS, MzR, LR	5
ClaR, MzS, LR	7
Total	57

Cla: clarithromycin. Mz: metronidazole. L: levofloxacin. S: susceptible. R: resistant.

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
