# Peer review of "Antimicrobial Efficacy of Five Probiotic Strains Against Helicobacter pylori"

_antibiotics, 2020, doi:10.3390/antibiotics9050244_

Round 1
Reviewer 1 Report
Authors demonstrated bactericidal effects of 5 commercially available probiotic strains and their conditioned media on 57 Helicobacter pylori strains. The number of H. pylori strains tested is really impressive. But bactericidal activity of Lactobacillus spp and other probiotic strains is well-known. So, the demonstration of such activity for strains that are commercially used as probiotics and therefore were tested for this activity does not provide any novel information. Moreover, the paper is quite badly written. The Introduction gives a very formal information about probiotics. The results section has obvious inaccuracies: Table 1 provides intervals for values instead of standard mean+-SD; strain categorization (Table 2) according antibiotic resistance is not accompanied supporting information how oarticular resistance mechanisms could influence on interactions with probiotic strains; Fig. 4 does not include statistical information, and any explanation of different resistance between Hp groups is not suggested; Fig. 6- in the Results section it is described as co-incubation experiment, but in Materials and Methods the experiment on incubation of Hp with media conditioned with probiotic strains is described. The conclusion about applicability of the probiotic strain together with antibiotics is not supported by anything. The only conclusion could be made from the experiments presented is that the tested probiotic strains and the media conditioned with these strains provide bactericidal effects against Hp.
Author Response
Dear Colleague,
thank you for your helpful suggestions.
Please see the following point by point answers:
Authors demonstrated bactericidal effects of 5 commercially available probiotic strains and their conditioned media on 57 Helicobacter pylori strains. The number of H. pylori strains tested is really impressive. But bactericidal activity of Lactobacillus spp and other probiotic strains is well-known. So, the demonstration of such activity for strains that are commercially used as probiotics and therefore were tested for this activity does not provide any novel information. Moreover, the paper is quite badly written. The Introduction gives a very formal information about probiotics. The results section has obvious inaccuracies:
Table 1 provides intervals for values instead of standard mean+-SD
R: We thank the reviewer for the comments. Table 1 has been corrected as suggested.
strain categorization (Table 2) according antibiotic resistance is not accompanied supporting information how particular resistance mechanisms could influence on interactions with probiotic strains. That was not our intention.
R: Antibiotic resistance mechanisms of H. pylori are not fully understood. The known ones should have nothing to do with the mechanisms that make H. pylori susceptible to the antimicrobial action of probiotics (at least in vitro: low pH and inhibition of urease activity). But, because we had data on resistance patterns available, we thought it would be wiser to divide the statistics by “H. pylori resistance patterns”, in order to observe if any changes in the efficacy of probiotics were detectable. This in vitro study confirmed that the inhibiting activity of probiotics is not related in any way to the antibiotic resistance of the H.pylori strain. We have better clarified this issue in the text. Row 228-232
Fig. 4 does not include statistical information, and any explanation of different resistance between Hp groups is not suggested;
R: Fig 4 (now fig 2) has been corrected as suggested
Fig. 6- in the Results section it is described as co-incubation experiment, but in Materials and Methods the experiment on incubation of Hp with media conditioned with probiotic strains is described.
R: Clarified in fig. 6 (now fig 4).
The conclusion about applicability of the probiotic strain together with antibiotics is not supported by anything.
R: You are absolutely right. We concluded indeed that probiotic antimicrobial activity on H. pylori does not depend on H. pylori antibiotic resistance patterns. Probiotic strains were selected for their resistance to low pH and bile salts, and their growth characteristics; but a clinical trial is needed. Row 342-344
The only conclusion could be made from the experiments presented is that the tested probiotic strains and the media conditioned with these strains provide bactericidal effects against Hp.
R: Once more, you are absolutely right.
Reviewer 2 Report
Given the challenges associated with (H. pylori) HP treatment, research into new therapeutic approaches is strongly warranted. This paper evaluates the anti-HP activity of 5 probiotic strains against H pylori. The results are interesting and potentially important. The manuscript could be improved by addressing the following points:
- Provide some more introductory and descriptive detail in the results section about the Tables and Figures presented.
- Provide error bars for the IZ means presented in Fig 3 and Fig 4 (which should be labelled Figs 1 and 2, respectively)
- Indicate statistically significant results on all graphs where appropriate
-
Provide information on how the clinical isolates of HP were obtained. Was the study approved by a research ethics committee? Was patient consent obtained to culture clinical HP isolates?
Author Response
Dear Colleague,
thank you for your helpful suggestions.
Please see the following point by point answers:
Given the challenges associated with (H. pylori) HP treatment, research into new therapeutic approaches is strongly warranted. This paper evaluates the anti-HP activity of 5 probiotic strains against H pylori. The results are interesting and potentially important. The manuscript could be improved by addressing the following points:
- Provide some more introductory and descriptive detail in the results section about the Tables and Figures presented.
- R: We thank the reviewer for the comments. Tables have been corrected as suggested.
- Provide error bars for the IZ means presented in Fig 3 and Fig 4 (which should be labelled Figs 1 and 2, respectively)
- R: We thank the reviewer for the comments. Figures have been corrected as suggested
- Indicate statistically significant results on all graphs where appropriate.
- R: text was amended as suggested
- Provide information on how the clinical isolates of HP were obtained. Was the study approved by a research ethics committee? Was patient consent obtained to culture clinical HP isolates?
- R: In our unit we always perform culture and drug susceptibility test for H. pylori positive patients as a routine, to prescribe chemosusceptibility-tailored therapies. These strains were from patients who signed an informed consent to participate to the study, approved by the Ethical committee.
Reviewer 3 Report
The aim of this study was to assess in vitro the bacteriostatic and bactericidal activity of probiotic strains (Lactobacillus casei DGDG ,Lactobacillus paracasei LPC-S01, Lactobacillus acidophilus LA14, Bifidobacterium lactis BL04, Streptococcus thermophilus ST21) against H. pylori, as a preliminary experiment to plan a clinical trial.
The article is well written and easy to read.
I suggest to the authors to shorten the abstract a little. In my opinion, the data are presented too broadly.
In my opinion, the introduction part is quite short. The authors could present more recent literature data specific to this topic.
The results are presented in a logical order. In the discussion, the authors compare their own results with those in the literature.
I suggest, however, that the results obtained from statistical interpretations of the data obtained in experiments should also be noted in graphs.
Finally, authors should present some important data in the conclusions.
Author Response
Dear Colleague,
thank you for your helpful suggestions.
Please see the following point by point answers:
The aim of this study was to assess in vitro the bacteriostatic and bactericidal activity of probiotic strains (Lactobacillus casei DGDG ,Lactobacillus paracasei LPC-S01, Lactobacillus acidophilus LA14, Bifidobacterium lactis BL04, Streptococcus thermophilus ST21) against H. pylori, as a preliminary experiment to plan a clinical trial.
The article is well written and easy to read.
I suggest to the authors to shorten the abstract a little. In my opinion, the data are presented too broadly.
- R: We thank the reviewer for the comments. Abstract has been corrected as suggestedR: introduction has been corrected as suggested row 59-62R: graphs have been amended as suggestedR: conclusions have been amended as suggested row 341-343
- Finally, authors should present some important data in the conclusions.
- The results are presented in a logical order. In the discussion, the authors compare their own results with those in the literature.I suggest, however, that the results obtained from statistical interpretations of the data obtained in experiments should also be noted in graphs.
- In my opinion, the introduction part is quite short. The authors could present more recent literature data specific to this topic.
Round 2
Reviewer 1 Report
The authors considerably improved the manuscript.